# Hear it here: Built environments predict ratings and descriptions of ambiguous sounds

**Brandon J. Forys[1]\*, Emily Qi[1], Rebecca M. Todd[1,2], Alan Kingstone[1]**

**1** Department of Psychology, University of British Columbia, Vancouver, BC, Canada, **2** Djavad Mowafaghian Centre for Brain Health, University of British Columbia, Vancouver, BC, Canada

\* brandon.forys@psych.ubc.ca

**Data Availability Statement:** The data is available on OSF at https://osf.io/wysdb/, DOI: 10.17605/OSF.IO/WYSDB.

**Funding:** Natural Sciences and Engineering Research Council of Canada, 170077-2011, Alan

## Abstract

The built environments we move through are a filter for the stimuli we experience. If we are in a darker or a lighter room or space, a neutrally valenced sound could be perceived as more unpleasant or more pleasant. Past research suggests a role for the layout and lighting of a space in impacting how stimuli are rated, especially on bipolar valence scales. However, we do not know how affective experiences and descriptions of everyday auditory stimuli are impacted by built environments. In this study, we examine whether listening to a series of ambiguously valenced sounds in an older, darker building leads these sounds to be rated as less pleasant—and described using more negatively valenced language—compared to listening to these sounds in a newer building with more natural light. In a between-subjects design, undergraduate participants at an older building or a newer building ($n_{Old}$ = 46, $n_{New}$ = 46; $n_{Female}$ = 71, $n_{Male}$ = 18, $M_{Age}$ = 21.18, $Range_{Age}$ = 17-38) listened to ten sounds that had previously been rated as ambiguous in valence, then rated these sounds on a bipolar valence scale before being asked to describe, in writing, how they felt about each sound. Participants rated sounds as being more pleasant at the New site compared to the Old site, but the sentiment of their descriptions only differed between sites when controlling for collinearity. However, bipolar scale ratings and description sentiment were highly correlated. Our findings suggest a role for the features of built environments in impacting how we appraise the valence of everyday sounds.

## Introduction

The built environments we inhabit in our daily lives affect how we perceive the world around us. If you are in a dark building with narrow hallways and hear a dog barking, you may feel more threatened than if you heard a dog barking in a bright, airy space—but why is this the case? Stimuli are specific sensory experiences that can guide our decisions in how we interpret the global environment in which they are situated. Auditory stimuli, unlike many kinds of visual stimuli, are especially susceptible to misinterpretation because it is generally more difficult to localize the source of auditory stimuli [1]. Furthermore, the perceived valence of a

Kingstone Natural Sciences and Engineering Research Council of Canada, F19-05182, Rebecca M. Todd Natural Sciences and Engineering Research Council of Canada, 569604, Brandon J. Forys.

**Competing interests:** The authors have declared that no competing interests exist.

sound can change depending on whether the person experiencing the sound can localize [2] and control [3] the sound.

If the valences of simultaneous auditory and visual stimuli differ, cross-modal effects can result in the perceived valence of one stimulus type priming the experienced valence of the other stimulus type [4, 5]—even if we are not selectively attending to one of the stimuli [6]. This transfer of valence between stimuli has previously been characterized through the Affect Misattribution Procedure (AMP) [7], an experimental paradigm capturing the misattribution of affective information based on contextual information [8]. Thus, in a space with unwelcoming features, we may perceive a neutral or even pleasant sound as unpleasant [9] compared to if we heard that sound in a space with pleasant features. While past research has established cross-modal priming given the co-presentation of two stimuli with differing valences, we do not yet know whether we would see priming from more global environmental features to a specific stimulus.

The features of a built environment beyond the lab—especially its architectural features, such as the size of a room or the amount of lighting available—can promote specific behaviours or cognitive states that differ from what can be characterized in the lab. We can form a unified representation of this environment that modulates how we experience stimuli and make decisions within it [10]. The cognitive ethology approach [11–13] evaluates how our ability to complete various tasks is impacted by these tasks' context. By focusing on how stimuli are situated and interpreted in their specific environments, we can better understand how the environment influences our perception of these stimuli. To this end, past studies have identified ways in which spaces with distinct features can nudge us towards making specific behavioural decisions through subtle incentives and cues [14, 15]. For example, Wu et al. [16] found that being in a building that promoted environmental sustainability increased the likelihood of participants selecting the correct bin in which to correctly recycle products. In the domain of auditory stimulus valence, Tajadura-Jiménez et al. [17] found specific effects of architectural features on sound ratings, such that participants rated sounds heard behind them in smaller rooms as more pleasant than those heard in front of them in larger rooms—but not if the sounds were threatening. Similar effects can be ascribed to outdoors spaces. In a study where participants rated the valence of an outdoor soundscape on a university campus, they reported the soundscape as more positively valenced if people were not speaking loudly and traffic noise was not heard [18]. Additionally, a fountain noise was rated as being negatively valenced—in spite of the fountain's pleasant appearance—as it compounded the negative experience of other unpleasant sounds. These studies exemplify how even subtle environmental features—such as architecture linked to sustainable design, or traffic noise heard in a courtyard—can shift people's behaviours and change how sets of stimuli within these environments are interpreted. However, they do not examine whether stimuli with valences open to interpretation are differentially appraised depending on these environmental features.

These features can manifest in myriad ways depending on the architectural configuration of a space. For example, we might expect that a room in an older building with no natural lighting may be less pleasant compared to a room in a new building with plentiful natural lighting. Such an older building might be characterized by cramped spaces and insufficient lighting, making it more difficult to know how to interpret stimuli in the space and how they might affect us. Thus, we might respond to stimuli in the less pleasant space more negatively. These features could change how we determine the sources of auditory stimuli [19] and whether we find them threatening [20]. No one element of these environments drives the behaviour. The general representation of the built environment could impact how we appraise stimuli within it through a process of behavioural priming, such that an environment with more positively valenced features could prime positive stimulus ratings—with negative

features priming negative ratings [4]. Past studies primarily focus on how features of the built environment shift behaviours, or how stimuli in the environment impact appraisals of one's context. However, they do not capture how environmental features impact appraisals of specific stimuli that do not have a strong initial valence. Furthermore, existing studies typically rely on limited survey or scale-based measures of responses to these stimuli, whereas our experiences of these stimuli may draw on a variety of features that cannot be captured by these traditional measures alone.

Experiences of stimuli could also differ greatly depending on whether we are in a laboratory context or a more naturalistic setting [11]. Design features of a space, including lighting, room size, and materials used, can also reduce the unpleasantness of auditory stimuli with negatively valenced features, driving generalization of stimulus valence from the laboratory to naturalistic contexts [17]. Accordingly, the ways in which participants respond to ambiguous stimuli could differ greatly if they are in a laboratory context vs. a more naturalistic setting—and whether they experience them in a highly controlled space vs. in built environments with differing acoustic and lighting properties. However, we do not know the extent to which the effects of context on perceived stimulus valence are reflected in participants' experiences of these stimuli. Such experiences could be captured by asking participants to describe how they feel about the stimuli they experience in each space, taking a phenomenological approach [21] to understanding how sounds may be differentially experienced depending on the features of the built environment in which they are heard.

The ways in which we interpret a sound could also be influenced not only by the sound's present spatial environment, but also by our understanding of what the sound means to us. Hearing a certain sound that we associate with a negative experience may evoke a strong emotional response regardless of present context and may, in fact, evoke a past spatial and temporal context instead [22]. Additionally, our perception of an ambiguous sound could be an indicator for how we interpret the environment in which it is heard [23]. Thus, it is important to evaluate how past and present experiences of sounds impact how we rate and perceive them. Valence rating scales allow participants to rate the degree to which these sounds elicit positive or negative sentiments, offering important information about quantifiable negative or positive aspects of an experience. However, they may not fully capture qualitative experiences of sounds in different built environments. Furthermore, we do not yet know how experiences of auditory stimuli are reflected in the context in which they are presented, or in descriptions of features and affective states associated with these stimuli.

In the present study, we take an ethological approach [11, 12] to explore whether common stimuli are contextually rated and described as positive or negative in different architectural environments. Participants completed the study at one of two different sites—a windowless room in an older building and a windowed room in a newer building. In a windowless room, participants have less of a reference for the outside world compared to a room with windows, which could cause the space—and stimuli experienced within it—to be judged as less pleasant. Furthermore, if one has to walk through an older building to reach this study room, one will experience more antiquated infrastructure that might be more effortful to traverse (i.e. narrower staircases or a smaller elevator), potentially driving a more negative appraisal of stimuli experienced in that space as compared to in a newer building. Thus, these two spaces offer contrasts in the environments that could drive stimulus interpretations. At each site, participants listened to a series of everyday sounds that have previously been rated as neutral with a high standard deviation on a bipolar scale, indicating varied representations of valence [24]. Such sounds would not have acoustic content (i.e. frequencies) or semantic-affective associations that would make them strongly pleasant or painful to listen to by themselves [25], allowing their perceived valence to be shifted by the built environment in which they are heard.

Furthermore, these sounds would be open to interpretation based on contexts in which they were previously experienced, and negatively or positively valenced experiences associated with these past contexts [20]. Participants first rated each sound's valence on a bipolar scale, which has been shown to disentangle valence from arousal information in emotionally salient stimuli more effectively than unipolar scales for each valence dimension [26]. These sound ratings were used to establish a self-report baseline for how participants rated the valence of each sound in different built environments. Then, to explore how participants describe their experiences of each sound, we asked them to write out how they felt about each sound. Their responses could then be evaluated for sentiment information—whether they used more positive or negative language to describe sounds depending on the context [27]. Last, they indicated the route they used to enter the building where the study was located, to evaluate participants' awareness of the features of their built environment. This allowed us to evaluate the role of movement through architectural spaces in impacting how participants experience ambiguous sounds.

## Materials and methods

### Participants

We powered our study for a medium effect of 0.4 for a difference in bipolar sound ratings between sites, based on the effect sizes observed in an online pilot study comparing ratings on unipolar vs. bipolar scales with the same stimulus set as used in the present study. This effect size is also congruent with that found by Tajadura-Jiménez et al. [17] when looking at the effect of architectural features and sound position on perceived sound valence. Using a bootstrapped power analysis for 1000 iterations in the Superpower package in R [28], we obtained a target sample size of $N = 64$ per site with a power level of 0.8. We recruited $N = 97$ psychology undergraduate participants from the Human Subject Pool at the University of British Columbia, between March 11th, 2022 and February 21st, 2024. All participants gave written, informed consent, and received bonus points towards their courses. Of these, $N = 5$ participants were unable to finish the task because of equipment errors. Therefore, we analyzed data from $N = 92$ participants ($n = 18$ male, $n = 71$ female, $n = 3$ other; Table 1). This study was approved by the research ethics board at the University of British Columbia with code H10-00527.

### Stimulus presentation

All visual stimuli were presented on a Lenovo ThinkPad P52s laptop with a 15.6 inch display (resolution: 1920x1080) running Windows 10, via the Pavlovia online study platform using PsychoPy 2021.2.3 (RRID: SCR_006571) [29]. Auditory stimuli were presented via the task through a set of two Dell A215 stereo speakers. In order to create the impression that the auditory stimuli originated from the room as opposed to from the computer—and was situated within the architectural context—the left channel speaker was placed directly behind the

**Table 1. Demographic information for all participants, by site and sex.**

| Site | Sex | n | $M_{age}$ | $SD_{age}$ | $Min_{age}$ | $Max_{age}$ |
|---|---|---|---|---|---|---|
| New | Female | 35 | 21.91 | 4.09 | 18 | 38 |
| | Male | 10 | 20.70 | 1.7 | 18 | 24 |
| | Other | 1 | 21.00 | | 21 | 21 |
| Old | Female | 36 | 20.63 | 3.34 | 17 | 36 |
| | Male | 8 | 21.62 | 1.69 | 19 | 24 |
| | Other | 2 | 19.00 | 0 | 19 | 19 |

**Table 2. Summary data for all IADS-2 sounds used, from original IADS study and from the current study.** IADS ratings have been centred from an original range of 1-10 to a range of -5 to 5, to match the scale used in the study. IADS-2: International Affective Digitized Sounds.

| Sound ID | Description | $M_{IADSrating}$ | $SD_{IADSrating}$ | $M_{IADSarousal}$ | $SD_{IADSarousal}$ | $M_{rating}$ | $SD_{rating}$ |
|---|---|---|---|---|---|---|---|
| 107 | Dog | 0.47 | 2.22 | 5.85 | 2.22 | 0.04 | 1.43 |
| 109 | Carousel | 1.40 | 2.13 | 5.64 | 2.13 | -0.04 | 1.73 |
| 113 | Cows | 0.45 | 1.71 | 4.88 | 1.71 | 0.11 | 1.31 |
| 114 | Cattle | 0.01 | 1.85 | 6.04 | 1.85 | -0.45 | 1.54 |
| 120 | Rooster | 0.20 | 2.10 | 5.41 | 2.10 | 0.32 | 1.55 |
| 152 | Tropical | 0.23 | 2.28 | 5.51 | 2.28 | 0.58 | 1.78 |
| 170 | Night | 0.31 | 2.12 | 4.60 | 2.12 | 0.02 | 2.00 |
| 171 | CountryNight | 0.59 | 1.79 | 3.71 | 1.79 | 0.17 | 1.65 |
| 262 | Yawn | 0.26 | 1.58 | 2.88 | 1.58 | -0.23 | 1.38 |
| 322 | TypeWriter | 0.01 | 1.82 | 4.79 | 1.82 | -0.57 | 1.48 |
| 361 | Restaurant | 0.36 | 1.62 | 5.01 | 1.62 | -0.21 | 1.50 |
| 373 | Paint | 0.09 | 1.55 | 4.65 | 1.55 | -0.22 | 1.41 |
| 374 | Sink | 0.60 | 1.35 | 4.23 | 1.35 | 0.29 | 1.47 |
| 378 | Doorbell | 1.06 | 2.01 | 6.15 | 2.01 | -0.23 | 1.63 |
| 403 | Helicopter1 | 0.57 | 1.83 | 5.56 | 1.83 | -0.70 | 1.29 |
| 425 | Train | 0.09 | 1.42 | 5.15 | 1.42 | 0.44 | 1.47 |
| 602 | Thunderstorm | 0.99 | 2.23 | 3.77 | 2.23 | 0.99 | 2.00 |
| 698 | Rain2 | 0.18 | 1.94 | 4.12 | 1.94 | 0.67 | 1.94 |
| 704 | Phone1 | 0.49 | 1.98 | 6.54 | 1.98 | -1.40 | 1.41 |
| 724 | Chewing | 0.34 | 1.97 | 4.91 | 1.97 | -0.90 | 1.97 |

participant, while the right channel speaker was placed directly to the right of the participant. The volume on these speakers was set to maximum, while the computer volume was set to a level of 32 in Windows.

## Stimuli

The stimuli in our study were a series of 20 audio recordings of everyday sounds, each with a uniform original duration of 6000 ms (Table 2). We selected these auditory stimuli from the International Affective Digitized Sounds system (IADS-2; [24]). These sounds were presented at the same volume with no fading in or out. All sounds selected had a mean rating between 5 (neutral point) and 5.9 on a 9-point Self-Assessment Manikin scale rating from completely unhappy to completely happy [24], and all but one sound ("Sink") had a standard deviation (SD) in their pleasure rating of at least 1.5. These sounds were played in two different random orders twice through the task: once for participants to rate how they felt about the sound on an onscreen bipolar scale, and again for participants to describe how they felt about the sound.

## Procedure

Participants came to the study room in-person to complete the task. In order to provide two distinct architectural environments in which the sounds would be heard, the study took place in two locations on the campus of the University of British Columbia (Table 3). In the old building condition (Old; Fig 1A), participants completed the study in a 2.00m x 3.00m room in a 1980s-era building on campus with a fully covered window and no natural lighting. In the new building condition (New; Fig 1B), participants completed the study in a 2.00m x 3.00m room in a 2010s-era building on campus with two large windows and plentiful natural lighting.

**Table 3. Information about each site used in the study.**

| Site | Room dimensions (m) | Site construction date | Room features |
|---|---|---|---|
| Old | 2.00 x 3.00 | 1984 | Windowless |
| New | 2.00 x 3.00 | 2013 | Windowed |

The laptop and speakers used, as well as the orientation of the participant relative to the door in the room, were kept consistent between sites. In both conditions, the participant was seated at a desk in front of the laptop. After reviewing the consent form and filling out demographic information and a questionnaire about COVID-related stress, participants began the experimental task. In the first block of the task (Fig 2A), participants listened to a series of ten sounds presented in a random order. After hearing each sound, participants were instructed to rate the sound on how pleasant or unpleasant it was on a scale from -4 (most unpleasant) to 4 (most pleasant), with a neutral point at 0. In the second block of the task (Fig 2B), participants heard the same ten sounds in a differently randomized order and were instructed to describe how they felt about each sound in 1-2 sentences by typing in an on-screen text box. Participants were not reminded of their sound ratings in this stage. Afterwards, participants were sent to a debrief survey in which they were asked to indicate how pleasant the room they were currently in was, on a 7-item Likert scale from "Very unpleasant" to "Very pleasant" with a neutral point. They were then asked how familiar they were with the building they were currently in, on a 7-item Likert scale from "Not familar at all" to "Very familiar" with a neutral point. They were further asked which entrance they used to enter the building (north, south, or side entrance) and, once in the building, whether they used the stairs or elevator to reach the study room. Participants then received credit towards their psychology courses.

## Analyses

We conducted all analyses using R 4.4.2 "Pile of Leaves" [30] through RStudio [31].

Our primary predictor variables was site—that is, whether the participant completed the study in an older, windowless room in a darker, concrete building (Old) or a newer, windowed room in a brighter, more modern building (New). Our secondary predictors were room

A
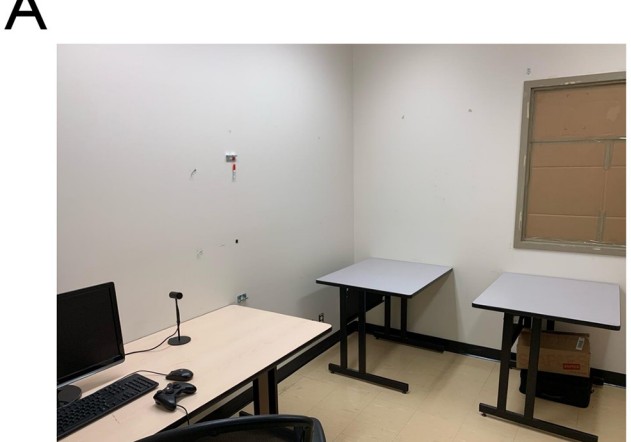

B
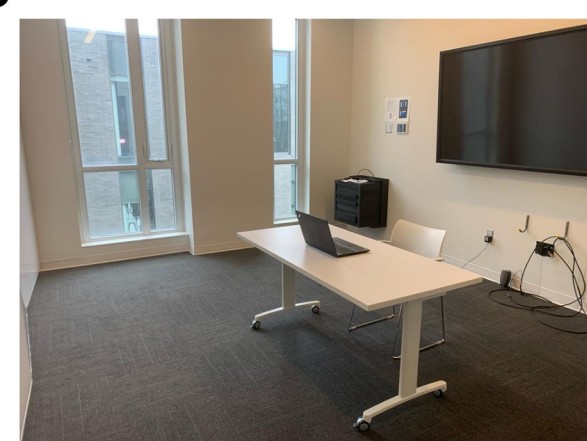

**Fig 1. Images of the sites used in each condition.** (A) The site used in the Old condition; (B) the site used in the New condition.

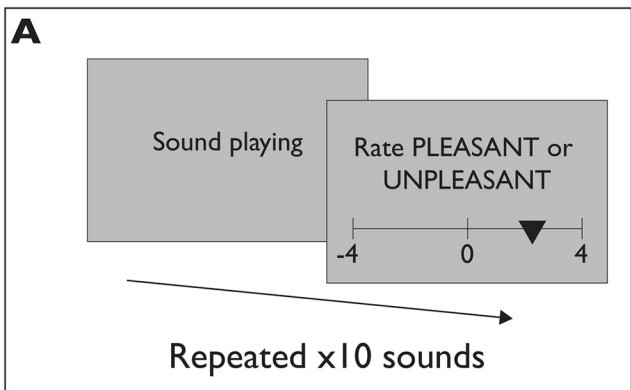
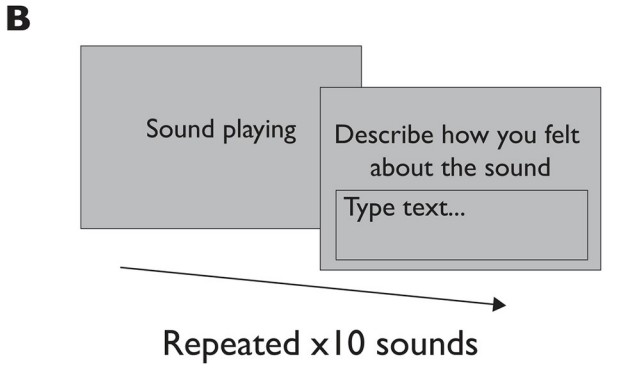

**Fig 2. Diagram of the trials in each phase of the task.** In the first phase (A), participants listen to and rate the valence of 20 sounds on a bipolar scale. In the second phase (B), participants listen to the same 20 sounds and describe in text how they feel about the sound.

pleasantness, operationalized as the participant's rating of the room's pleasantness; site familiarity, operationalized as the participant's familiarity with the building in which they were completing the study; entrance used, operationalized as the entrance participants used to arrive at the study; and path, operationalized as whether participants used the elevator or stairs to arrive at the study. Our primary dependent variables were 1) bipolar scale rating, operationalized as the participant's rating of each sound's valence on a bipolar scale; and 2) sentiment score, operationalized as the sentiment of each body of text that participants used to describe how they felt about each sound. We calculated sentiment scores by running the VADER sentiment analysis R package [27] on each body of text that participants wrote, and obtaining the overall sentiment of each sentence. VADER calculates sentiment scores by evaluating the valence of each word in each text response—accounting for the order of, and modifiers on, each word—to output a compound sentiment score for each participant's response to each sound.

We compared bipolar scale and sentiment analysis ratings by site using two mixed-effects linear models using the lmerTest package in R [32] predicting bipolar ratings and sentiment scores, respectively, from each site with participant and sound used as random effects. We followed these analyses with a multi-level model predicting bipolar ratings and sentiment scores from rating type (bipolar rating vs. sentiment score), site, room pleasantness, site familiarity, entrance used, and path as fixed effects and participant as a random effect (illustrated in Fig 3). We further evaluated the correlation between bipolar ratings and sentiment score to explore whether both scales captured information about perceptions of sound valence. Additionally, to control for collinearity in bipolar scale and sentiment score results, we conducted a multivariate ANOVA using the manova function in R [33] with valence ratings and sentiment scores as distinct dependent variables.

## Results

We first asked whether participants rated sounds as being significantly more pleasant on the bipolar scale at the New site relative to the Old site, to see if built environment can impact how participants rate ambiguously valenced sounds on a conventional self-report scale. A linear mixed-effects model evaluating sound ratings by site, controlling for sound as a random effect, revealed that participants rated all sounds as being significantly more pleasant in the New site compared to the Old site (Table 4; Fig 4A).

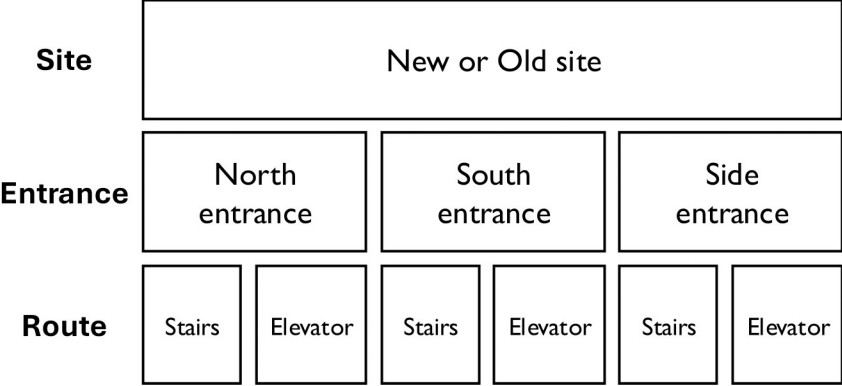

**Fig 3. Schematic of the factor structure of the statistical analyses used.**

We next evaluated whether the sentiment of participants' descriptions of how they rated sounds was significantly more positively valenced at the New site compared to the Old site, to determine whether built environment shifted the content of unprompted descriptions of sounds by participants. A linear mixed-effects model evaluating sound sentiment scores by site, controlling for sound as a random effect, revealed that the sentiment of sound descriptions did not significantly differ between sites (Table 5; Fig 4B).

The participant experience of each site may have differed on a variety of factors, including their route to the study room and their familiarity with the site itself, which could impact the built environment in which sounds were experienced. We therefore conducted a multi-level model analysis to investigate whether experiences of the built environment directly explained

**Table 4. Linear mixed effects analysis results for sound ratings by sound and site.** AIC = Akaike information criterion, BIC = Bayesian information criterion, ICC = intraclass correlation, RMSE = root mean squared error.

|  | Multi-level model: Rating |
|---|---|
| (Intercept) | 0.14 |
|  | (0.15) |
| Site | -0.40** |
|  | (0.13) |
| SD (Intercept participant) | 0.54 |
| SD (Intercept sound) | 0.54 |
| SD (Observations) | 1.51 |
| Num.Obs. | 1822 |
| R2 Marg. | 0.014 |
| R2 Cond. | 0.217 |
| AIC | 6844.1 |
| BIC | 6871.6 |
| ICC | 0.2 |
| RMSE | 1.47 |

+ $p < 0.1$,

* $p < 0.05$,

** $p < 0.01$,

*** $p < 0.001$

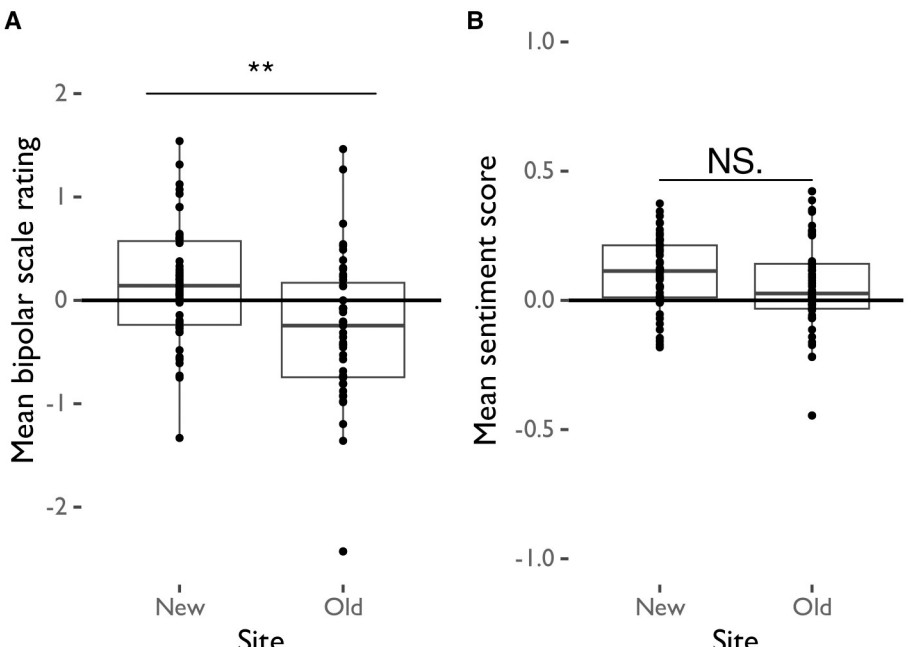

**Fig 4. Mean sound ratings from (A) bipolar scale and (B) sentiment analysis scores by site.**

sound ratings and sentiment scores. This analysis revealed that site, rating scale type, and whether the south vs. north entrance to the building was used significantly predicted sound ratings (Table 6; Fig 5) such that bipolar sound ratings, but not sentiment scores, were lower in the Old site compared to the New site and participants who used the south entrance to both

**Table 5. Linear mixed effects analysis results for sentiment scores by sound and site.** AIC = Akaike information criterion, BIC = Bayesian information criterion, ICC = intraclass correlation, RMSE = root mean squared error.

|  | Multi-level model: Sentiment |
|---|---|
| (Intercept) | 0.10* |
|  | (0.04) |
| Site | -0.04 |
|  | (0.03) |
| SD (Intercept participant) | 0.11 |
| SD (Intercept sound) | 0.16 |
| SD (Observations) | 0.50 |
| Num.Obs. | 1820 |
| R2 Marg. | 0.001 |
| R2 Cond. | 0.136 |
| AIC | 2739.4 |
| BIC | 2766.9 |
| ICC | 0.1 |
| RMSE | 0.49 |

+ p < 0.1,

* p < 0.05,

** p < 0.01,

*** p < 0.001

**Table 6. Multi-level model analysis results for sound ratings.** AIC = Akaike information criterion, BIC = Bayesian information criterion, ICC = intraclass correlation, RMSE = root mean squared error.

|  | Multi-level model: Rating |
|---|---|
| (Intercept) | 0.09 |
|  | (0.19) |
| Pleasantness of room | -0.01 |
|  | 1 (0.03) |
| Familiarity with building | -0.02 |
|  | (0.03) |
| Side entrance used | 0.11 |
|  | 2 (0.10) |
| South entrance used | 0.22* |
|  | (0.11) |
| Stairs vs. elevator | 0.07 |
|  | 1 (0.10) |
| Rating type | 0.13* |
|  | (0.06) |
| Site | -0.23* |
|  | (0.10) |
| SD (Intercept participant) | 0.24 |
| SD (Observations) | 0.41 |
| Num.Obs. | 184 |
| R2 Marg. | 0.102 |
| R2 Cond. | 0.338 |
| AIC | 282.8 |
| BIC | 315.0 |
| ICC | 0.3 |
| RMSE | 0.36 |

+ p < 0.1,

* p < 0.05,

** p < 0.01,

*** p < 0.001

sites generally rated and described the sounds as being more pleasant. Room pleasantness, site familiarity, whether the side entrance to the building was used, and whether stairs or an elevator were used did not significantly predict sound ratings. Given that the site influenced ratings but not the valence of descriptions, we further examined the degree to which ratings and sentiments were correlated. A correlation analysis revealed that participant ratings on bipolar scales were significantly correlated with the sentiment of their descriptions of the sounds ($t(92) = 8.31$, $p < 0.001$, $r = 0.65$). Furthermore, a multivariate ANOVA analysis predicting bipolar scale ratings and sentiment scores revealed that, taken together, both measures significantly differed by site and by sound (Site: $F(2, 1785) = 16.51$, $p < 0.001$, *Pillai* = 0.02; Sound: $F(38, 3572) = 7.37$, $p < 0.001$, *Pillai* = 0.15).

## Discussion

We investigated whether the architectural environment in which an ambiguously valenced sound is heard influences how participants rate the valence of the sound and the sentiment of

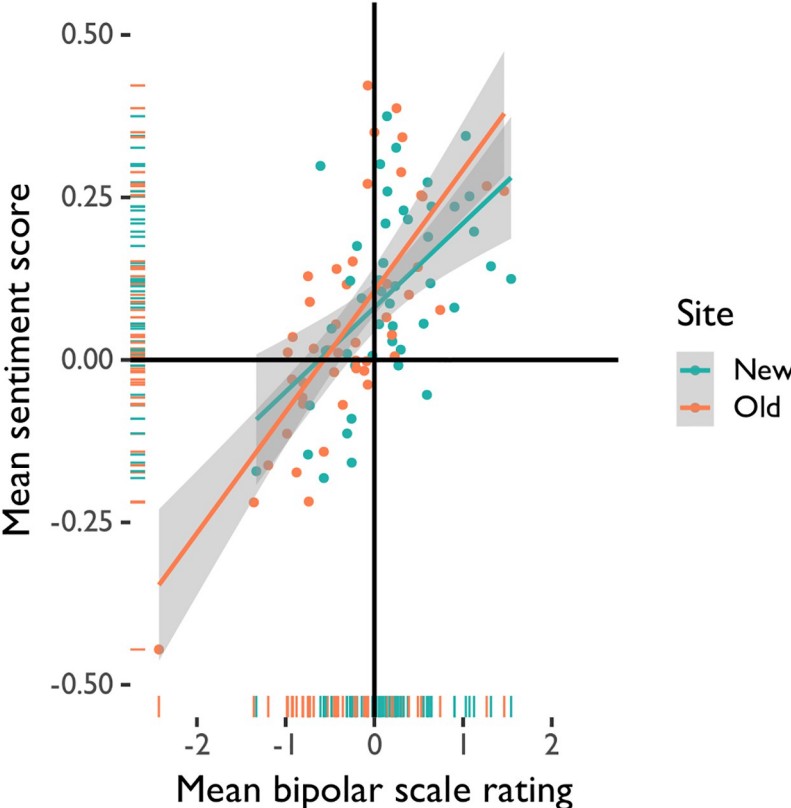

**Fig 5. Linear regression of mean sound ratings from bipolar scale and sentiment analysis scores by site.**

how they describe it. We found that, in an older building with a windowless, artificially lit room (Old), participants rated all sounds as being significantly less pleasant than did participants in a newer building with a naturally lit room (New). When participants described how they felt about the sounds, the valence of the sentiment of these descriptions only differed between sites in our multivariate ANOVA analysis—which controlled for collinearity—but not in our linear mixed model analyses. Participants thus used moderately more negatively valenced language to describe sounds at the older site compared to the newer site.

Our findings align with past work showing that qualities of the built environment—such as the features and design of a building—can impact how we approach stimuli and make decisions—including findings by Tajadura-Jiménez et al. [17] showing that sounds were rated as more negatively valenced in smaller compared to larger rooms. The sites for our study did not have explicit cues meant to nudge specific behaviours [14, 15]—there was no signage or interventions that promoted more positive or negative appraisals of sounds. Nevertheless, differences between sites still had a significant effect on how participants rated sounds as negative or positive. Each site had distinct architectural, acoustic, and affective features, which may have impacted the decisions that participants make about stimuli they experience in each site.

Participants may have offloaded some of the cognitive effort required to make a judgment about a sound to how they interpreted its context [34]. In the Old site, participants navigated a predominantly concrete building to reach a study room that was highly enclosed and had no natural lighting. In contrast, to reach the naturally lit study room in the New site, participants navigated a bright and airy space. The less pleasant Old site may thus have drawn attention to

the more negative attributes or associations of an otherwise neutral, everyday sound, with the New site comparatively enhancing the positive aspects of sounds heard. This is consistent with enactive views in cognitive science [35], which highlight the notion that cognition is a process of valenced sensemaking, tracking events that are either good or bad for the organism's well-being in continuous engagement with a complex environment.

Further to this point, participants rated and described sounds more positively when using the south compared to the north entrance of the building. The south entrance of each building was generally further away from the test site than the north entrance. Participants entering the building by this route may have experienced more of the built environment that would impact their ratings of the sounds. Information that is encoded about architectural features could differ depending on the entrance participants use to reach the study. Through the cognitive ethology approach [11] adopted in this study, we thus show the impacts of naturalistic experiences of the environment around us on interpretations of everyday stimuli.

As shown in the AMP [9], this information can prime attributions of the valence that participants assign to stimuli they experience in the space—even if participants are unaware of the effects of the space on their sound ratings and descriptions. Thus, participant ratings of sounds could be proxies for their experience of the architectural context in which the sounds were heard—enhanced by the amount of time they spent moving through the space. It is important to note that—contrary to earlier findings—recent evaluations of the AMP suggest that cross-modal priming is dependent on our awareness of being primed [36]. However, as participant perceptions of room pleasantness were not linked to sound ratings, it is reasonable to speculate that the effects observed in the present study were implicitly primed. Our findings highlight potential latent roles for the combined features of an architectural context in impacting not only behaviours but also experiences of stimuli.

The attribution of differences in sound interpretation to the site in which they were heard may be challenged by the lack of difference in sentiment ratings between sites in participant descriptions of sounds in our linear mixed model analyses. In contrast, the bipolar rating scale invites interpretations of a sound on an axis of negative to positive valence. These scales have been shown to be dissociable from arousal [26], while participant descriptions of their experiences of sounds could range much more widely and be more nuanced. As the sounds used were not strongly positively or negatively valenced and had a wide range of ratings when originally normed [24] (2), it is not surprising that this wide range of experiences would be reflected in how they were described. In spite of our finding that only bipolar sound ratings differed between sites given linear mixed model analyses, sentiment scores were significantly correlated with bipolar ratings—a result supported by our multivariate ANOVA analysis showing that site predicted both sound ratings and sentiment scores. This MANOVA result's contrast with the non-significant linear mixed model results could speak to how the linear mixed models do not account for collinearity among bipolar ratings and sentiment scores, thus weakening the power of this analysis to detect differences between aspects of the auditory experience by site as compared to the MANOVA. The participant descriptions—from which sentiment scores were derived—captured a wider variety of experiences than valence information; however, the valence information that was present was congruent with that of bipolar ratings. This finding highlights the importance of using both quantitative and qualitative research methods in complementary ways—as each method can capture different aspects of how a built environment can impact perceptions of ambiguous sounds.

Further to this point, many participants drew on their memories of past experiences with the sounds, beyond merely describing how they experienced the sound in their present context. For example, one of the sounds was of a dog barking. While some participants described the sound outside of any temporal context (e.g., "This sound is not that pleasant because it

feels like the dog is sensing danger", others drew on past experiences with dogs to inform their present appraisal of the experience (e.g. "[I] quite like this one. it sounds like what i would hear from my childhood bedroom."). Other sounds had ambiguous sources and meanings to different participants. For example, one sound used sounded like nighttime in a rainforest to some participants ("This sound feels like home, during nighttime at a tropical country, it could be soothing to me."); to others, it sounded like an unpleasant dental tool ("[This] sound makes me uncomfortable. it reminds me of the drying device used at the dentist. [The] sound makes my mouth and skin feel uncomfortably dry listening to it."). Participants did not reference the architectural context in which they heard the sounds at any point. However, this diverse array of participant responses reflects the variety of reasons that participants may have had for rating sounds as more or less pleasant—ratings that did interact with the architectural context in which they were presented. Queries about the qualitative experiences of stimuli in our environment augment information from self-report scales and offer rich phenomenological information [21] about whether participants situate these stimuli in their present context or, instead, draw on past contexts and experiences, when interpreting how they feel about these stimuli.

We must consider a number of qualifications when interpreting these results. First, because of the nature of the sites used, their room configuration could not be kept perfectly consistent. In the Old site, although the participant and speaker positions were held constant across all participants, the arrangement of other furniture and objects in the room underwent minor changes over the course of the study (e.g., a chair was pulled out rather than tucked under a desk). However, these changes did not affect the lighting, space available, or layout in the room, and the qualitative impression of the space was maintained throughout the study. Similarly, in the New site, the furniture stayed relatively constant; however, changes in natural lighting throughout the day during different runs of the study may have impacted the room's character. Furthermore, for $n = 24$ participants, the room partition on one wall in the New site was locked in an open position, causing other parts of the room to be visible. However, the rest of the space was the same in appearance to this study room. Second, in both sites, loud conversations and sounds outside the room were reported by $n = 5$ participants. However, such experiences were typical of these sites and did not interrupt the study. Third, the sounds used were highly open to interpretation in terms of valence. Although this was by design, it may have weakened potential effects of site compared to using sounds that had an established positive or negative valence—especially as previous studies have investigated reductions in the aversiveness of negatively valenced sounds depending on properties of the built environment [2].

Last, the order of the sound ratings and descriptions were not counterbalanced across participants. Participant descriptions of sounds may therefore have been informed by their memory of how they rated the sounds. However, the lack of a significant difference in the sentiment of sound ratings between sites—even when sound ratings on the bipolar scale significantly differed—suggests that the qualitative component of the study still captured distinct information about participants' sound experience beyond merely recapitulating the sound ratings. On the other hand, the robustness of the site difference in sound ratings in spite of these environmental differences speaks to how the overall feel of an architectural space can strongly drive perceptions of stimuli experienced within that space, and highlights the insights that can be derived from evaluating how stimuli are interpreted in real-world, uncontrolled environments.

Future studies could investigate whether and how individual attributes of a built environment drive quantitative and qualitative evaluations of participant experiences of sounds. For example, increasing the level of illuminance in a room or on a computer screen could increase the perceived positivity of a sound's valence [37], while reducing one's ability to attribute a

sound to a source could increase the negativity of its valence [2, 38]. Furthermore, given the increasing relevance of virtual and augmented reality in our everyday lives, future work should also explore similarities and differences in how sounds are rated and described in virtual vs. real-life spaces [39]. These projects would help us better understand whether and how we interpret everyday experiences beyond the computer screen and beyond the lab, in more naturalistic settings. These results could also have implications for a wide variety of fields, including designing theatre sound effects and music to elicit specific emotional responses in audiences and maximize the impact of a performance [40, 41].

Our study's findings emphasize the importance of considering the role of the architectural environment when we judge the valence and meaning of everyday sounds. In two buildings that frequently serve as experiment testing spaces, we show significant differences in how sounds are rated while controlling for sound identity as well as familiarity and perceived pleasantness of the built environment. Furthermore, our findings suggest a potential role for past experiences interacting with present context when participants give rich descriptions of their experiences of everyday sounds. Experiences of stimuli within architectural contexts could tell us about how the contexts themselves are perceived, shedding light on the role of features like lighting and acoustic qualities in determining how comfortable we feel in real-world spaces—as well as in how we can create artistic experiences that capture our imagination and inspire us. These results could inform future conversations on how to create built environments that are amenable to pleasant experiences, and emphasize the important of context when interpreting stimuli both inside and outside the lab.

## Author Contributions

**Conceptualization:** Brandon J. Forys, Rebecca M. Todd, Alan Kingstone.

**Data curation:** Brandon J. Forys.

**Formal analysis:** Brandon J. Forys, Emily Qi.

**Funding acquisition:** Rebecca M. Todd, Alan Kingstone.

**Investigation:** Brandon J. Forys, Emily Qi.

**Methodology:** Brandon J. Forys, Emily Qi, Rebecca M. Todd, Alan Kingstone.

**Project administration:** Alan Kingstone.

**Resources:** Alan Kingstone.

**Software:** Brandon J. Forys.

**Supervision:** Rebecca M. Todd, Alan Kingstone.

**Validation:** Brandon J. Forys.

**Visualization:** Brandon J. Forys.

**Writing – original draft:** Brandon J. Forys.

**Writing – review & editing:** Brandon J. Forys, Rebecca M. Todd, Alan Kingstone.

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
