## [Decision Letter · Decision Letter 0]

28 Aug 2024

PONE-D-24-27494Hear it here: Built environments predict ratings and descriptions of ambiguous soundsPLOS ONE

Dear Dr. Forys,

Thank you for submitting your manuscript to PLOS ONE. After careful consideration, we feel that it has merit but does not fully meet PLOS ONE’s publication criteria as it currently stands. Therefore, we invite you to submit a revised version of the manuscript that addresses the points raised during the review process.

Two experts in the field have carefully reviewed the interesting manuscript entitled “Hear it here: Built environments predict ratings and descriptions of ambiguous sounds“. You can find their comments below. 

On my side, I would like also to have means and std of the ratings to see if the stimuli have remained neutral with wide dispersion or not.

In light of these reviews and my own reading, I am requesting a major revision and resubmission, in which you will need to respond to each point in each review. 

We look forward to receiving your revised manuscript.

Kind regards,

Bruno Alejandro Mesz, Ph.D.

Academic Editor

PLOS ONE

“BJF: NSERC Canada Graduate Scholarship - Doctoral (CGS D) (569604)

AK: NSERC (170077-2011)

RMT: NSERC (F19-05182)”

Reviewers' comments:

Reviewer's Responses to Questions

**Comments to the Author**

1. Is the manuscript technically sound, and do the data support the conclusions?

Reviewer #1: Yes

Reviewer #2: Partly

2. Has the statistical analysis been performed appropriately and rigorously? 

Reviewer #1: Yes

Reviewer #2: No

3. Have the authors made all data underlying the findings in their manuscript fully available?

Reviewer #1: Yes

Reviewer #2: Yes

4. Is the manuscript presented in an intelligible fashion and written in standard English?

Reviewer #1: Yes

Reviewer #2: No

5. Review Comments to the Author

Reviewer #1: It would be interesting, perhaps for a future publication, to have a more detailed description of the audio signals used as stimuli for the study.

Furthermore, some parts of the article repeat concepts already developed.

Reviewer #2: The submitted manuscript addresses an interesting question: does the immediate (built) environment (i.e., a modern vs an old room) influence valence judgements about neutral auditory stimuli (drawn from the IADS stimulus set). The authors consider additionally some qualitative data which is subject to sentiment analysis, and nested independent variables such as route taken to the room. Although, in a broad sense, the study is adequately put together and reaches a logical conclusion, there are a number of reasons why the study is not publishable at present. Below I present some quite detailed comments - I hope that they are helpful to the authors in formulating future iterations of the manuscript for resubmission to PLOS or elsewhere.

Keywords: embodied cognition is mentioned only once in the manuscript and isn't really used as the conceptual framework here. Valence scales and stimulus ratings aren't particularly informative. Also, this paper does not particularly take an ethological approach.

Introduction:

The purpose of the article is clear. However, the literature review draws on all sorts of sources, and I think it could be more focused on environmental influences on auditory experience. There are some priming studies that consider cross-modal relationships between visual stimuli and sounds, e.g. Goerlich et al (2012), and there are plenty of others. https://pubmed.ncbi.nlm.nih.gov/22360592/) Tadajua-Jiminez (2010) https://psycnet.apa.org/buy/2010-09991-010, D'Alessandro et al (2018) https://journals.sagepub.com/doi/abs/10.1177/1351010X18778759 might also be useful in the literature review. It will be really important to build a rationale for the study here - there is useful material in auditory perception, soundscapes and architecture literature - as it is, some of the points seem a bit tangential.

If this phenomonen is conceptualised as transfer of affect from a valenced prime to a neutral target, the Affect Misattribution Procedure is probably the place to look for the cognitive underpinning of the effect - I was surprised not to see this mentioned, especially as priming is mentioned, albeit briefly, in the discussion.

Also, there are some instances where the authors cite something but don't capture accurately what the cited article is saying, e.g., Son et al. (2015) isn't an auditory study and doesn't directly bear on the present manuscript.

As with the keywords, the text leans on embodied cognition in the first paragraph, and claims to take an ethological approach later on - neither of these are really reflective of the manuscript as a whole.

There are some phrases that seem a little odd to me in academic writing, for instance "as such" is used four times. Additionally, there seems to be a mixture of registers in the writing. Most readers are happy with 'passive' scientific writing or with first person writing, but it would be better to stick with one or the other rather than mix.

There is a lot of explanation of the benefits of rating scales, and the some variation phrase 'bipolar rating scales' is used several times. This reads more like a student trying to convince a marker that they know what they are doing, rather than something written for an interested and scientifically literate audience.

Methods:

Line 85 - there is more to sounds than frequencies. It really is the semantic-affective content of the IADS sounds that is important. Psychoacoustics (and frequencies are not the only part of this) is important as a determinant of affect, but not the only thing.

More details of the stimuli would be good - which IADS sounds did you use? what were the valence/arousal ratings in the original Bradley et al study. Were they uniformly 6000 ms or did you allow variation to allow for ecological validity of the sounds. What about volume? Were the onsets/offsets ramped/faded or did the sounds just start/stop?

The main manipulation is the sites. Perhaps it would be worth including details of the sites in the method somewhere, e.g. a table comparing the two sites. Optionally, you could perhaps include photographs of the two sites?

Line 104 - 0.4 seems like a sensible effect size to base power analysis on, but how did you arrive at it? Is it based on previous research?

Overall, the procedure seems sound. Perhaps a schematic of the route*site*elevator design might be easier to digest than the text for some readers to understand the design quickly.

Results:

I have some concerns about parts of the analysis. A simple t-test between old and new sites is not quite the right thing here. There needs to be some control for how participants are using the scale, and I can't see how the t-test can handle the different IADS sounds. One option is to normalise the ratings. An even better option would be something like a linear mixed effects model with rating~site+(1|Participant)+(1|Sound), which would be easy enough to implement in R (eg using lmer).

Line 175 - a simple outline of how VADER calculates sentiment would be helpful to some readers, as there are a few different ways to calculate sentiment. From memory, the VADER GitHub has a neat description.

line 184: Bonferroni corrected for multiple comparisons - I can't really see where the multiple comparisons are. There are basically two separate t-tests.

Also, given that there are two correlated dependent variables, might manova remove some of the confusion?

Incidentally, the headings of the subsections refer to ratings but include analysis of the sentiments as well. I think the results section is short enough to be delineated by paragraphs rather the subsections.

Discussion:

'Nudge' is quite a specific concept in behavioural psychology. This should be introduced earlier or not at all, and discussed more fully if you use it. Same goes for 'priming' (line 252).

Line 233/236 - the switch to 'aversive' from negative seems like you're talking about something slightly different from before. It would be easier for the reader if you were consistent in terminology.

As with the Introduction, the Discussion hops around a lot of areas. I'd suggest keeping it really focused on environmental/architectural influences on affective ratings of auditory stimuli. This has implications for instance in theatre design, and is interesting as a question in basic science.

Finally, I hope the authors are not disheartened by the review - this is an interesting manuscript, and has the potential to contribute to architectural/auditory psychology.

6. PLOS authors have the option to publish the peer review history of their article (what does this mean?). If published, this will include your full peer review and any attached files.

Reviewer #1: No

Reviewer #2: No

---

## [Author Response · Author response to Decision Letter 0]

9 Oct 2024

We would like to thank the editor and reviewers for their thoughtful and constructive comments and their positive assessment of our manuscript. We have responded to all the suggestions, which we itemize below. 

As a brief overview, our responses to the reviewers' comments involve several important additions. We strengthened the theoretical grounding of the Introduction, especially with regards to cognitive ethology and the priming literature. In the Methods section, we provided more details on the previous and current ratings of the auditory stimuli used. In the Results section, we conducted two mixed-effects models to control for the random effects of individual sounds, as well as a multivariate ANOVA to simultaneously predict bipolar ratings and sentiment scores. 

Our responses to Editor and reviewer comments are marked as “Response:”; changes in the paper are in bold face.

Editor:

Two experts in the field have carefully reviewed the interesting manuscript entitled “Hear it here: Built environments predict ratings and descriptions of ambiguous sounds“. You can find their comments below. 

On my side, I would like also to have means and std of the ratings to see if the stimuli have remained neutral with wide dispersion or not.

Response: Thank you. We have now added a table (please see Table 2) with the means and SDs of our study’s bipolar scale ratings, alongside the original IADS valence and arousal ratings. These results show that, in our study, sounds were rated similarly close to the neutral point of the scale, and had a slightly smaller range of dispersion, compared to how they were rated in the original IADS norming of these sounds.

In light of these reviews and my own reading, I am requesting a major revision and resubmission, in which you will need to respond to each point in each review. 

Response: We have now revised the manuscript’s formatting and naming scheme to match these style requirements.

“BJF: NSERC Canada Graduate Scholarship - Doctoral (CGS D) (569604)

AK: NSERC (170077-2011)

RMT: NSERC (F19-05182)”

Response: The funders had no role in study design, data collection and analysis, decision to publish, or preparation of the manuscript. We now state this in the revised cover letter along with a correction to AK's grant number: NSERC (2022-03079).

Reviewers' comments:

Reviewer #1: It would be interesting, perhaps for a future publication, to have a more detailed description of the audio signals used as stimuli for the study.

Furthermore, some parts of the article repeat concepts already developed.

Response: We appreciate this comment. We have now more thoroughly discussed the features of the auditory stimuli used for the study on page 6:

“The stimuli in our study were a series of 20 audio recordings of everyday sounds, each with a uniform original duration of 6000 ms (Table 2). We selected these auditory stimuli from the International Affective Digitized Sounds system (IADS-2; [24]). These sounds were presented at the same volume with no fading in or out. All sounds selected had a mean rating between 5 and 5.9 (neutral point) on a 9-point Self-Assessment Manikin scale rating from completely unhappy to completely happy [24], and all but one sound ("Sink") had a standard deviation (SD) in their pleasure rating of at least 1.5.”

We have also indicated the specific sounds used from the IADS stimulus set, as well as the valence and arousal ratings provided in the original IADS study, in Table 2. In general, we have revised the manuscript to reduce repetition.

Reviewer #2: The submitted manuscript addresses an interesting question: does the immediate (built) environment (i.e., a modern vs an old room) influence valence judgements about neutral auditory stimuli (drawn from the IADS stimulus set). The authors consider additionally some qualitative data which is subject to sentiment analysis, and nested independent variables such as route taken to the room. Although, in a broad sense, the study is adequately put together and reaches a logical conclusion, there are a number of reasons why the study is not publishable at present. Below I present some quite detailed comments - I hope that they are helpful to the authors in formulating future iterations of the manuscript for resubmission to PLOS or elsewhere.

Response: We thank the reviewer for their thoughtful comments and feedback, which we have sought to address below.

Keywords: embodied cognition is mentioned only once in the manuscript and isn't really used as the conceptual framework here. Valence scales and stimulus ratings aren't particularly informative. Also, this paper does not particularly take an ethological approach.

Response: We have now revised our choice of keywords to better reflect the themes discussed in the paper, as follows: “Auditory psychology”, “Architectural psychology”, “Auditory perception”, and “Priming”.

Introduction:

The purpose of the article is clear. However, the literature review draws on all sorts of sources, and I think it could be more focused on environmental influences on auditory experience. There are some priming studies that consider cross-modal relationships between visual stimuli and sounds, e.g. Goerlich et al (2012), and there are plenty of others. Tadajua-Jiminez (2010), D'Alessandro et al (2018) might also be useful in the literature review. It will be really important to build a rationale for the study here - there is useful material in auditory perception, soundscapes and architecture literature - as it is, some of the points seem a bit tangential.

Response: We have now revised the Introduction section (pages 2-5) to build a stronger rationale for environmental influences on auditory experience; cited and discussed the recommended priming studies as well as additional studies to support this rationale; and restructured the literature review such that it is more focused and cohesive.

If this phenomonen is conceptualised as transfer of affect from a valenced prime to a neutral target, the Affect Misattribution Procedure is probably the place to look for the cognitive underpinning of the effect - I was surprised not to see this mentioned, especially as priming is mentioned, albeit briefly, in the discussion.

Response: We now discuss the Affect Misattribution Procedure, together with a more in-depth discussion of behavioural priming and transfer of valence from the environment to the auditory stimuli, in the Introduction on pages 2-3:

“This transfer of valence between stimuli has previously been characterized through the Affect Misattribution Procedure (AMP) [7], an experimental paradigm capturing the misattribution of affective information to stimuli based on contextual information [8]. Thus, in a space with unwelcoming features, we may perceive a neutral or even pleasant sound as unpleasant [9] compared to if we heard that sound in a space with pleasant features. While past research has established cross-modal priming given the co-presentation of two stimuli with differing valences, we do not yet know whether we would see priming from more global environmental features to a specific stimulus.”

 We also discuss the AMP in the Discussion on pages 13-14:

“As shown in the AMP [9], this information can prime attributions of the valence that participants assign to stimuli they experience in the space - even if participants are unaware of the effects of the space on their sound ratings and descriptions. Thus, participant ratings of sounds could be proxies for their experience of the architectural context in which the sounds were heard - enhanced by the amount of time they spent moving through the space. It is important to note that - contrary to earlier findings - recent evaluations of the AMP suggest that cross-modal priming is dependent on our awareness of being primed [36]. However, as participant perceptions of room pleasantness were not linked to sound ratings, it is reasonable to speculate that the effects observed in the present study were implicitly primed.”

Also, there are some instances where the authors cite something but don't capture accurately what the cited article is saying, e.g., Son et al. (2015) isn't an auditory study and doesn't directly bear on the present manuscript.

Response: If the reviewer is referring to our citation of Song et al. (2019), we have replaced references to this paper with more appropriate, auditory-focused references to Tajadura-Jiménez et al. (2010) and Wardhani et al. (2022) in the Introduction on page 3 and in the Discussion on page 15, respectively.

As with the keywords, the text leans on embodied cognition in the first paragraph, and claims to take an ethological approach later on - neither of these are really reflective of the manuscript as a whole.

Response: We have revised the Introduction to remove the focus on embodied cognition in the first paragraph on page 2.

We have also introduced the ethological literature earlier and more thoroughly, on page 3:

“The cognitive ethology approach [11–13] evaluates how our ability to complete various tasks is impacted by these tasks’ context. By focusing on how stimuli are situated and interpreted in their specific environments, we can better understand how the environment influences our perception of these stimuli.”

There are some phrases that seem a little odd to me in academic writing, for instance "as such" is used four times. Additionally, there seems to be a mixture of registers in the writing. Most readers are happy with 'passive' scientific writing or with first person writing, but it would be better to stick with one or the other rather than mix.

Response: We thank the reviewer for their feedback; we have revised the Introduction section to increase the variety of phrases used and revised the overall paper to use a consistent first-person register.

There is a lot of explanation of the benefits of rating scales, and the some variation phrase 'bipolar rating scales' is used several times. This reads more like a student trying to convince a marker that they know what they are doing, rather than something written for an interested and scientifically literate audience.

Response: We have now revised the discussion of rating scales on pages 3-4 to be more concise and less repetitive to improve readability. 

Methods:

Line 85 - there is more to sounds than frequencies. It really is the semantic-affective content of the IADS sounds that is important. Psychoacoustics (and frequencies are not the only part of this) is important as a determinant of affect, but not the only thing.

More details of the stimuli would be good - which IADS sounds did you use? what were the valence/arousal ratings in the original Bradley et al study. Were they uniformly 6000 ms or did you allow variation to allow for ecological validity of the sounds. What about volume? Were the onsets/offsets ramped/faded or did the sounds just start/stop?

Response: We have now added additional information about the auditory stimuli used in the study. In Table 2, we list all sounds used and the original valence and arousal ratings reported for each sound from the IADS study. Furthermore, on page 6, we have now revised our description of the sounds to be more precise:

“The stimuli in our study were a series of 20 audio recordings of everyday sounds, each with a uniform original duration of 6000 ms (Table 2). We selected these auditory stimuli from the International Affective Digitized Sounds system (IADS-2; [24]). These sounds were presented at the same volume with no fading in or out. All sounds selected had a mean rating between 5 and 5.9 (neutral point) on a 9-point Self-Assessment Manikin scale rating from completely unhappy to completely happy [24], and all but one sound ("Sink") had a standard deviation (SD) in their pleasure rating of at least 1.5.”

Additionally, we now mention in the Introduction on page 5 that the sounds did not have strong initial semantic-affective associations.

The main manipulation is the sites. Perhaps it would be worth including details of the sites in the method somewhere, e.g. a table comparing the two sites. Optionally, you could perhaps include photographs of the two sites?

Response: In Table 3, we have now included details on the spatial configuration and characteristics of each site. In Figure 1, we have now included pictures of the study room at each site.

Line 104 - 0.4 seems like a sensible effect size to base power analysis on, but how did you arrive at it? Is it based on previous research?

Response: We thank the reviewer for their feedback. On page 5, we have now clarified that this effect size was selected based on results from an online pilot study focusing on sound ratings for unipolar vs. bipolar scales, and are congruent with those observed in an existing study comparing sound ratings:

“We powered our study for a medium effect of 0.4 for a difference in bipolar sound ratings between sites, based on the effect sizes observed in an online pilot study comparing ratings on unipolar vs. bipolar scales with the same stimulus set used in the present study. This effect size is also congruent with that found by Tajadura-Jiménez et al. [17] when looking at the effect of architectural features and sound position on perceived sound valence.”

Overall, the procedure seems sound. Perhaps a schematic of the route*site*elevator design might be easier to digest than the text for some readers to understand the design quickly.

Response: We thank the reviewer for their comments and have now included a schematic of the design in Figure 3.

Results:

I have some concerns about parts of the analysis. A simple t-test between old and new sites is not quite the right thing here. There needs to be some control for how participants are using the scale, and I can't see how the t-test can handle the different IADS sounds. One option is to normalise the ratings. An even better option would be something like a linear mixed effects model with rating~site+(1|Participant)+(1|Sound), which would be easy enough to implement in R (eg using lmer).

Response: We have revised the Results section to replace the t-tests with the recommended linear mixed effects models (separated for predictors of sound ratings and sentiment scores), the results for which we now report in Tables 4 and 5, and described as follows on page 9: 

“A linear mixed-effects model evaluating sound ratings by site, controlling for sound as a random effect, revealed that participants rated all sounds as being significantly more pleasant in the New site compared to the Old site (Table 4, Figure 4A).”

“A linear mixed-effects model evaluating sound sentiment scores by site, controlling for sound as a random effect, revealed that the sentiment of sound descriptions did not significantly differ between sites (Table 5, Figure 4B).”

We have also updated the Methods section on page 8 to introduce this method:

“We compared bipolar scale and sentiment analysis ratings by site using two mixed-effects linear models using the lmerTest package in R [32] predicting bipolar ratings and sentiment scores, respectively, from each site with participant and sound used as random effects.”

Line 175 - a simple outline of how VADER calculates sentiment would be helpful to some readers, as there are a few dif

---

## [Decision Letter · Decision Letter 1]

3 Dec 2024

PONE-D-24-27494R1Hear it here: Built environments predict ratings and descriptions of ambiguous soundsPLOS ONE

Dear Dr. Forys,

Thank you for submitting your manuscript to PLOS ONE. After careful consideration, we feel that it has merit but does not fully meet PLOS ONE’s publication criteria as it currently stands. Therefore, we invite you to submit a revised version of the manuscript that addresses the points raised during the review process.

**Dear authors, thanks for your careful attention to reviewers' comments; I find the manuscript has greatly improved. **

**There is still one point that is confusing for me: linear mixed models and MANOVA give different results with respect to the significance of sentiment description differences between both sites. However, in the abstract and discussion you state that there were no differences. Please clarify this choice.**

We look forward to receiving your revised manuscript.

Kind regards,

Bruno Alejandro Mesz, Ph.D.

Academic Editor

PLOS ONE

**Journal Requirements:**

Reviewers' comments:

Reviewer's Responses to Questions

**Comments to the Author**

1. If the authors have adequately addressed your comments raised in a previous round of review and you feel that this manuscript is now acceptable for publication, you may indicate that here to bypass the “Comments to the Author” section, enter your conflict of interest statement in the “Confidential to Editor” section, and submit your "Accept" recommendation.

Reviewer #1: All comments have been addressed

2. Is the manuscript technically sound, and do the data support the conclusions?

Reviewer #1: Yes

3. Has the statistical analysis been performed appropriately and rigorously? 

Reviewer #1: Yes

4. Have the authors made all data underlying the findings in their manuscript fully available?

Reviewer #1: Yes

5. Is the manuscript presented in an intelligible fashion and written in standard English?

Reviewer #1: Yes

6. Review Comments to the Author

**Reviewer #1:** (No Response)

7. PLOS authors have the option to publish the peer review history of their article (what does this mean?). If published, this will include your full peer review and any attached files.

Reviewer #1: No

---

## [Author Response · Author response to Decision Letter 1]

4 Dec 2024

We would like to once again thank the editor and reviewers for their thoughtful and constructive comments and recommendations to our manuscript. We have addressed this additional comment below and feel that the manuscript is stronger as a result of their feedback. 

In our response to the editor’s comments, we now discuss the differences in the results given by the linear mixed models and the MANOVA, with regards to potential differences in how well each model accounts for collinearity. Changes in the paper are in bold face.

We address the editor’s comment below.

Dear authors, thanks for your careful attention to reviewers' comments; I find the manuscript has greatly improved. 

There is still one point that is confusing for me: linear mixed models and MANOVA give different results with respect to the significance of sentiment description differences between both sites. However, in the abstract and discussion you state that there were no differences. Please clarify this choice.

Response:

We thank the editor for their suggestions. We have now revised the abstract and discussion section to explain the differences in the MANOVA and linear mixed model results. The linear mixed effect models did not account for potential collinearities in the regressors used, while the multivariate ANOVA was able to capture the contribution of site and sound identity to both bipolar ratings and sentiment levels. The changes in the abstract are as follows:

“Participants rated sounds as being more pleasant at the New site compared to the Old site, but the sentiment of their descriptions only differed between sites when controlling for collinearity”.

The changes in the discussion section are as follows. On page 10:

“When participants described how they felt about the sounds, the valence of the sentiment of these descriptions only differed between sites in our multivariate ANOVA analysis - which controlled for collinearity - but not in our linear mixed model analyses. Participants thus used moderately more negatively valenced language to describe sounds at the older site compared to the newer site.”

On pages 11-12:

“The attribution of differences in sound interpretation to the site in which they were heard may be challenged by the lack of difference in sentiment ratings between sites in participant descriptions of sounds in our linear mixed model analyses.”

“In spite of our finding that only bipolar sound ratings differed between sites with linear mixed model analyses, sentiment scores were significantly correlated with bipolar ratings - a result supported by our multivariate ANOVA analysis showing that site predicted both sound ratings and sentiment scores. This MANOVA result’s contrast with the non-significant linear mixed model results could speak to how the linear mixed models do not account for collinearity among bipolar ratings and sentiment scores, thus weakening the power of this analysis to detect differences between aspects of the auditory experience by site as compared to the MANOVA.”

---

## [Editor Report · Decision Letter 2]

8 Dec 2024

Hear it here: Built environments predict ratings and descriptions of ambiguous sounds

PONE-D-24-27494R2

Dear Dr. Forys,

We’re pleased to inform you that your manuscript has been judged scientifically suitable for publication and will be formally accepted for publication once it meets all outstanding technical requirements.

Kind regards,

Bruno Alejandro Mesz, Ph.D.

Academic Editor

PLOS ONE
---

## [Editor Report · Acceptance letter]

11 Dec 2024

PONE-D-24-27494R2 

PLOS ONE

Dear Dr. Forys, 

I'm pleased to inform you that your manuscript has been deemed suitable for publication in PLOS ONE. Congratulations! Your manuscript is now being handed over to our production team.

Kind regards, 

on behalf of

Dr. Bruno Alejandro Mesz 

Academic Editor

PLOS ONE